# Gradient Statistics-Based Multi-Objective Optimization in Physics-Informed Neural Networks

**DOI:** 10.3390/s23218665

**Published:** 2023-10-24

**Authors:** Sai Karthikeya Vemuri, Joachim Denzler

**Affiliations:** 1Computer Vision Group, Friedrich Schiller University Jena, 07743 Jena, Germany; joachim.denzler@uni-jena.de; 2German Aerospace Center, Institute of Data Science, 07743 Jena, Germany

**Keywords:** physics-informed neural networks, multi-objective optimization, loss weighting

## Abstract

Modeling and simulation of complex non-linear systems are essential in physics, engineering, and signal processing. Neural networks are widely regarded for such tasks due to their ability to learn complex representations from data. Training deep neural networks traditionally requires large amounts of data, which may not always be readily available for such systems. Contrarily, there is a large amount of domain knowledge in the form of mathematical models for the physics/behavior of such systems. A new class of neural networks called Physics-Informed Neural Networks (PINNs) has gained much attention recently as a paradigm for combining physics into neural networks. They have become a powerful tool for solving forward and inverse problems involving differential equations. A general framework of a PINN consists of a multi-layer perceptron that learns the solution of the partial differential equation (PDE) along with its boundary/initial conditions by minimizing a multi-objective loss function. This is formed by the sum of individual loss terms that penalize the output at different collocation points based on the differential equation and initial and boundary conditions. However, multiple loss terms arising from PDE residual and boundary conditions in PINNs pose a challenge in optimizing the overall loss function. This often leads to training failures and inaccurate results. We propose advanced gradient statistics-based weighting schemes for PINNs to address this challenge. These schemes utilize backpropagated gradient statistics of individual loss terms to appropriately scale and assign weights to each term, ensuring balanced training and meaningful solutions. In addition to the existing gradient statistics-based weighting schemes, we introduce kurtosis–standard deviation-based and combined mean and standard deviation-based schemes for approximating solutions of PDEs using PINNs. We provide a qualitative and quantitative comparison of these weighting schemes on 2D Poisson’s and Klein–Gordon’s equations, highlighting their effectiveness in improving PINN performance.

## 1. Introduction

Deep learning has witnessed unprecedented growth and continues to improve various fields, showcasing the immense potential that has yet to be fully explored. In recent years, it has made significant inroads into science and technology, enabling the understanding and modeling of complex systems. However, two key challenges with deep learning are its black-box nature and its requirement for vast data. Acquiring such extensive data may not always be feasible, and ensuring that the trained model comprehends the scientific task accurately is imperative. Fortunately, an opportunity arises from the abundance of domain knowledge available in the form of physical laws. Integrating this knowledge into deep learning approaches holds the promise of bridging the gap between data-driven learning and incorporating the physical laws that govern the systems. Such integration would not only enhance the capabilities of neural networks but also enable them to learn from data and fundamental scientific principles, thereby offering a powerful framework for advancing scientific research and technological applications. Physics-Informed Neural Networks (PINNs) provide an elegant framework to accomplish this integration. They combine *physics* in the form of partial differential equations of a system with classical neural networks used for data-driven solutions. PINNs were introduced by Lagaris et al. [1] and were popularized by seminal works [2,3]. In recent years, PINNs have garnered widespread attention in scientific machine learning for their demonstrated ability to use underlying physics [4]. The success of PINNs (and deep learning in general) can be attributed to the robust and simplified implementations of Automatic Differentiation (AD) frameworks [5]. These frameworks facilitate easy computations of a composite function’s gradients with respect to its parameters and inputs [6].

PINNs are widely used for solving differential equations’ forward and inverse problems (ordinary/partial/fractional/integro) [7,8]. For solving the forward problem of a partial differential equation (PDE), a neural network is approximated as the solution of the differential equation. It is common in the literature to formulate the loss function as a linear combination of the differential equation and its boundary and initial conditions. The neural network is trained to minimize the multi-object loss function. This leads to the training of the neural network towards the particular solution of the differential equation for the given boundary/initial conditions.

The use of PINNs has rapidly increased in recent years, with successful applications in various areas such as Computational Fluid Dynamics [9,10], Geosciences [11,12], Signal Processing [6,13,14], and Climate Sciences [15]. However, these networks have encountered limitations and problems. PINNs are found to be difficult to train to achieve correct solutions [16,17]. Specifically, the neural network fails to learn the correct solution, gives erroneous solutions, and the training becomes unstable. It is found that for certain scenarios, the neural network shows bias towards a wrong solution and even training for a larger number of epochs will not help to overcome this failure [16]. Manual scaling of loss terms or imposing hard boundary conditions [18] only works for a limited number of cases and it is not universally applicable. This difficulty is attributed to the imbalanced multi-objective optimization at the heart of PINNs. Recent works have been dedicated to discovering failure modes and techniques to achieve balanced training and enhanced performance. Enhancements were suggested for the architecture of the Neural Network used [17,19], activation functions [20], and the formulation of boundary conditions [6,21].

In this paper, we focus on enhancing the multi-objective loss function of a PINN. This is achieved by weighting individual loss terms to ensure proper training. By proper training, we mean that all the loss functions contribute in a way that the solution of the PINN converges towards the particular solution of the differential equation. The weights are updated dynamically during the training. The motivation for choosing these dynamically adaptive weights can come from different perspectives. In this study, we explore the potential of gradient statistics to define adaptive weighting strategies. We aim to leverage information from simple statistics (like mean, standard deviation, and kurtosis) of backpropagated gradients of individual loss terms during training and assign individual weights based on this information.

The schematic representation of implementing a weighting scheme in PINN training is shown in Figure 1. The blue box represents a feed-forward neural network whose input is the collocated space–time samples in the domain, boundaries, and initial condition. The output is the solution that needs to be approximated. Using AD, the derivatives of output with regard to input are estimated, and individual loss functions are formulated according to the definition and boundary/initial conditions. In the red box, the weights are estimated from the gradient statistics of individual loss terms before combining them into a single multi-objective loss function. This final loss function is used to train the PINN using an optimizer (generally Adam).

This line of work using gradient statistics for weighting is used in the prominent works of Wang et al. [17], who used the mean magnitude of the gradients to assign weights. In a similar line, Maddu et al. [22] used the standard deviation of gradients. The main contribution of this work is to extend them and introduce three novel schemes based on mean and standard deviation combinations and kurtosis along with the existing mean-based [17] and standard deviation-based [22] weighting schemes. Together with the original formulation of Raissi et al. [23], we evaluate these weighting schemes’ qualitative and quantitative performances by solving 2D Poisson’s and Klein–Gordon’s equations. In the literature, there have been other strategies for determining weights, like self-attention-based strategies [24] where the concept of self-attention is used to weigh individual loss terms, Neural Tangent Kernel (NTK)-based [16] weighting where the eigenvalues of the NTK matrix formed by individual loss terms under certain conditions are used for weighting different loss terms, etc. In this paper. we focus on gradient statistics.

In summary, our contributions are as follows:We give different perspectives from the literature that imbalanced training of multi-objective optimization is one of the main causes of failure of a classical PINN and emphasize the need to weigh individual loss terms to achieve balanced training.We propose that weighting the individual loss terms using backpropagated gradient statistics offers an elegant method to improve the training of PINNs.We formulate three novel weighting schemes based on combinations of mean and standard deviation and kurtosis and compare them to state-of-the-art mean [17] and standard deviation-based [22] schemes and show an improvement in the training of PINNs.

## 2. Theory

This section provides a basic overview of PINNs. We recommend referring to [2] for a more detailed overview.

### 2.1. Physics-Informed Neural Networks: Theory and Formulation

Consider a general differential equation written as
(1)$$u_{t}+D\left[u\right]=0,\;t\in \left[0,T\right],\;x\in \Omega ,$$
where ***u***(***x***,***t***) is the solution of the differential equation. In other words, the *physics* of quantity ***u*** in the region **Ω** within time ***T*** is described by Equation (Equation 1). **D[·]** represents the differential operator (linear or non-linear), which is a function containing derivatives with regard to ***x***∈Ω. $u_{t}$ represents the derivative of ***u*** with regard to the temporal coordinate. To resolve the behavior of ***u*** from (Equation 1), i.e., solve the differential equation, boundary conditions, and initial conditions are necessary. In other words, information about ***u*** along the boundary **∂Ω** and its initial state at time 0. The general formulation of boundary conditions (BCs) and initial conditions (ICs) is given by (Equation 2):(2)$$\begin{array}{ccc} u(0,x)=g(x), & x\in \Omega ,\\ B[u]=0, & t\in [0,T], & x\in \partial \Omega . \end{array}$$

The function g(x) is the initial state of the solution **u**. **B[·]** is the operator that defines the behavior at the boundary **∂Ω**. Following the formulation of Raissi et al. [2], a neural network, represented by $u_{\theta }$, with trainable parameters (weights and biases) **θ** approximates the latent, hidden solution ***u***. The inputs of this neural network are sampled points of time and space within and along the boundaries. A PINN aims to tune parameters **θ** such that the neural network approximates the solution of the differential equation. We calculate the required derivatives (of any order) of output ***u*** with regard to inputs ***x***, ***t*** by Automatic Differentiation. The differential equation provides information about how the combination of derivatives should *behave* at every point in the region Ω. Samples of input–output pairs are available at boundaries **∂Ω**. This model is trained on a loss function given by (Equation 3) with the individual terms given by (Equation 4)–(Equation 6).
(3)$$\;L\left(\theta \right)=L_{ic}\left(\theta \right)+L_{bc}\left(\theta \right)+L_{r}\left(\theta \right),$$
(4)$$L_{ic}\left(\theta \right)=\frac{1}{N_{ic}}\sum_{i=1}^{N_{ic}}{\left|u_{\theta }\left(0,x_{ic}^{i}\right)-g\left(x_{ic}^{i}\right)\right|}^{2},$$
(5)$$L_{bc}\left(\theta \right)=\frac{1}{N_{bc}}\sum_{i=1}^{N_{bc}}{\left|B\left[u_{\theta }\right]\left(t_{bc}^{i},x_{bc}^{i}\right)\right|}^{2},$$
(6)$$L_{r}\left(\theta \right)=\frac{1}{N_{r}}\sum_{i=1}^{N_{r}}{\left|\frac{\partial u_{\theta }}{\partial t}\left(t_{r}^{i},x_{r}^{i}\right)+D\left[u_{\theta }\right]\left(t_{r}^{i},x_{r}^{i}\right)\right|}^{2}.\;$$

The term $L_{r}$ is the loss term enforcing the differential equation, i.e., this term penalizes the points inside the region according to the differential operator. $L_{\mathrm{bc}},L_{\mathrm{ic}}$ are the loss terms that enforce the boundary and initial conditions of the differential equation, respectively. $x_{\mathrm{ic}},\left(t_{\mathrm{bc}},x_{\mathrm{bc}}\right)$, **and**
$\left(t_{r},x_{r}\right)$ are the initial, boundary, and residual (collocation points inside the domain) samples, $N_{\mathrm{ic}},N_{\mathrm{bc}}$, **and**
$N_{r}$ are the numbers of initial, boundary, and residual samples, respectively. The next step is to minimize these loss terms collectively using an optimizer that updates the parameters **θ** of the neural network. This makes the neural network output obey the differential equation and its boundary/initial conditions, thus approximating the solution for the given boundary and initial conditions. It is a multi-objective optimization problem where the *physics* is informed by the differential equation used to construct the loss function and initial and boundary data samples.

### 2.2. Failure of Multi-Objective Optimization of PINNs

Classical PINNs showed limitations in approximating solutions to several types of PDEs [16,17,22,25]. We give a brief overview of studies explaining the reasons for the failure of PINNs.

Wang et al. [17] studied the gradients being backpropagated from different loss terms. The work has shown that during training, some loss functions (mostly $L_{bc}$) tend to have backpropagated gradients becoming *stiff*, i.e., becoming zero/almost close to zero. This makes contributions of these loss terms insignificant during training. This unintended stiffness in gradient flow dynamics is given as failure of classical PINNs.Maddu et al. [22] showed that PINNs can be seen as a special case of Sobolev training. They suggested that loss terms containing higher-order derivatives, dominant high frequencies, or both (generally residual loss), exhibit dominance over other loss terms during training. This leads to non-existent or extremely weak gradients of remaining loss terms. They termed this phenomenon as *vanishing task-specific gradients*.Wang et al. [16] investigated when and why PINNs fail. They used the theory of Neural Tangent Kernels [26] to study the training dynamics of PINNs. They empirically showed that PINNs suffer from inherent spectral bias and have imbalanced convergence rates among different loss components. This discrepancy in convergence rates and spectral bias is attributed to the failure of PINNs.Rohrhofer et al. [25] showed that multi-objective optimization of PINNs heavily depends on its Pareto Front. They showed that for the classical formulation, the Pareto optimal has a significant loss value where the optimization step cannot be taken without increasing the individual loss values. They emphasized the need to scale loss terms to reach a better Pareto optimal.

## 3. Gradient-Based Dynamic Weighting Schemes for Effective Training of PINNs

Imbalanced training of the multi-objective loss function of the PINN leads to erroneous results. Hence, we opted for the approach of balancing them. The straightforward way for scaling a loss term is by multiplying it with a scalar *weight*, denoted by $\lambda_{i}$, where i∈r, ic, and ∈bc represent the weights for residual, initial, and boundary loss, respectively. Thus the original loss function in (Equation 3) can be rewritten as (Equation 7):(7)$$\;L\left(\theta \right)=\lambda_{ic}L_{ic}+\lambda_{bc}L_{bc}+\lambda_{r}L_{r},$$
where $\lambda_{\mathrm{ic}},\lambda_{\mathrm{bc}},\lambda_{r}$ are scalar weights for IC, BC, and residual, respectively.

Now, a weighting scheme aims to ensure that the imbalance explained in the previous section does not happen. As already mentioned in the previous section, what is happening is that during the training some loss terms are vanishing. They are not contributing during the gradient descent step of the optimizer (generally Adam), i.e., their gradients with regard to trainable parameters of the PINN are becoming zero. This phenomenon makes the solution converge to an erroneous solution. These erroneous solutions are shown as baselines. The weighting scheme should ensure that individual loss terms’ gradients do not become zero during training and help the optimizer reach the optima corresponding to the correct differential equation solution. Therefore, a weighting scheme should achieve the following things:Keep track of gradients of individual loss terms forming the multi-objective loss terms of a PINN (typically **$L_{r},L_{ic},L_{bc}$**) during training.Detect and update the individual weights to prevent gradients from falling to zero.

A gradient statistics-based weighting scheme uses empirical statistics for the aforementioned tasks. The idea is that by looking at the empirical statistics of the individual backpropagated gradients, we obtain information about the distribution, which is used both as a detector and an update formula. For example, Wang et al. [17] used the mean of magnitude of gradients as a statistic. If the mean magnitude of the distribution of gradients of a particular loss term falls to zero during training, it is a sign of stiffness/vanishing. This is avoided by using the inverse of mean magnitude as a weight to the loss terms. This way, whenever a loss term shows signs of stiffness, the weight is changed to train the PINN to the desired solution.

The main contribution of this paper is to present a collection of such gradient statistics-based weighting schemes. The first one is the aforementioned mean-based weighting scheme [17]. Maddu et al. [22] used the inverse of standard deviation to assign weights similarly. Building on these works, we introduce three novel weighting schemes based on mean and standard deviation combinations and kurtosis. Detailed formulations of all schemes are provided subsequently, but a short motivation is discussed here. The motivation behind formulating these schemes is that the combination of mean and standard deviation gives even more pronounced information about the distribution’s variability and central tendency, thus making the combination of the mean and standard deviation a better weighting scheme. We introduce two such combinations: the sum and product of mean and standard deviation. We also introduce kurtosis as a useful statistic because it captures the peakedness of gradients around the mean. Therefore, kurtosis and standard deviation are used together as a weighting scheme. Now, we describe all the weighting schemes (two taken from the literature and three novel).

We calculate the empirical mean, maximum, standard deviation, and kurtosis of backpropagated gradients at regular intervals of epochs during training. We follow the method described by [22] for calculating mean and standard deviation and adopt a similar approach for kurtosis calculation. Mean and maximum values are calculated for the absolute values of the gradients. Standard deviation and kurtosis are calculated for the original values.

### 3.1. Mean-Based Weighting Scheme

Wang et al. [17] introduced the first gradient-based weighting scheme based on the mean magnitude of gradients. The mean magnitude of gradients falling to zero indicates that the loss term is vanishing. Hence, using the inverse of the mean as a weight will penalize the loss term when the gradients fall to zero. The formula for determining weights is given by Equation (Equation 8):(8)$$\begin{array}{c} {\hat{\lambda }}_{k}\left(\tau \right)=\frac{\max\{|\nabla_{\theta^{\mathrm{sh}}}L_{r}\left(\tau \right)|\}}{|\bar{|\nabla_{\theta^{\mathrm{sh}}}L_{k}\left(\tau \right)|}|},\\ \lambda_{k}\left(\tau +1\right)=\alpha \lambda_{k}\left(\tau \right)+\left(1-\alpha \right){\hat{\lambda }}_{k}\left(\tau \right), \end{array}$$

Here, ${\hat{\lambda }}_{k}\left(\tau \right)$ denotes the calculated weight of **k**-th loss term at training epoch τ. The operator **$\theta^{\mathrm{sh}}$** denotes gradients with regard to parameters that are trainable (shared). The weight for the next epoch is updated as the moving average of the current weight (${\hat{\lambda }}_{k}\left(\tau \right)$) and the previous weight ($\lambda_{k}\left(\tau \right)$). The parameter controlling the moving averages is denoted by **α**. The denominator is the mean magnitude of gradients, and the numerator has the maximum value among the gradients coming from **$L_{r}$**. The parameter α is the controlling parameter of the updated weight, and its value is taken as 0.5. This will remain the same for all the weighting schemes.

### 3.2. Standard Deviation-Based Weighting Scheme

Maddu et al. [22] suggested that standard deviation is a more suited statistic. Standard deviation captures the spread or variability of the empirical distribution of gradients. A decreasing standard deviation value (for a constant mean = 0) indicates that the distribution is becoming narrower, and the values are falling closer to zero. This is a sign of stiffness and a vanishing task. Therefore, the inverse of standard deviation as weight balances the variability of all loss terms. The weights are given by
(9)$$\begin{array}{c} {\hat{\lambda }}_{k}\left(\tau \right)=\frac{\mathrm{std}\{\nabla_{\theta^{\mathrm{sh}}}L_{r}\left(\tau \right)\}}{\mathrm{std}\{\nabla_{\theta^{\mathrm{sh}}}L_{k}\left(\tau \right)\}}, \end{array}$$

The operator std{·} denotes the calculation of standard deviation. It calculates the standard deviation of the given distribution (here, gradients of a loss term). Building on these two approaches given by (Equation 8) and (Equation 9), we present two different weighting schemes based on the backpropagated gradient statistics of different loss terms.

### 3.3. Mean and Standard Deviation-Based Weighting Schemes

We propose the following extension of using mean and standard deviation jointly. Using the standard deviation and mean together can provide a balance between measuring the variability and the central tendency of the gradients. The standard deviation captures the spread or dispersion of the gradients across the data points. The mean captures their average or typical value, which may be informative about the overall behavior of the loss terms. Low mean and standard deviation could give a better indication of vanishing tasks. Therefore, using the inverse of a combination of mean and standard deviation could provide a better balance among the loss terms.

We consider two types of combinations: summation and multiplication, given by Equations (Equation 10) and (Equation 11), respectively:(10)$$\begin{array}{cc} {\hat{\lambda }}_{k}\left(\tau \right) & =\frac{\mathrm{std}\left\{\nabla_{\theta^{\mathrm{sh}}}L_{r}\left(\tau \right)\right\}+\max\{|\nabla_{\theta^{\mathrm{sh}}}L_{r}\left(\tau \right)\left|\right\}}{\mathrm{std}\left\{\nabla_{\theta^{\mathrm{sh}}}L_{k}\left(\tau \right)\right\}+\left|\bar{|\nabla_{\theta^{\mathrm{sh}}}L_{k}\left(\tau \right)|}\right|}, \end{array}$$
(11)$$\begin{array}{cc} {\hat{\lambda }}_{k}\left(\tau \right) & =\frac{\mathrm{std}\left\{\nabla_{\theta^{\mathrm{sh}}}L_{r}\left(\tau \right)\right\}\cdot |\bar{|\nabla_{\theta^{\mathrm{sh}}}L_{r}\left(\tau \right)|}|\bar{}}{\mathrm{std}\left\{\nabla_{\theta^{\mathrm{sh}}}L_{k}\left(\tau \right)\right\}\cdot \left|\bar{|\nabla_{\theta^{\mathrm{sh}}}L_{k}\left(\tau \right)|}\right|}. \end{array}$$

Using moving averages to update the weight for the next epoch remains the same as the previous strategy.

### 3.4. Kurtosis and Standard Deviation-Based

By considering the kurtosis and standard deviation, one can distinguish between different types of variability in the gradients. Kurtosis measures the peakedness or flatness of the distribution, in our case, of the gradients. This reveals whether they are concentrated around the mean or widely dispersed. High kurtosis (leptokurtic) indicates that values are peaked at the mean, as illustrated in the Figure 2. It shows that a high kurtosis can indicate that gradients peak around zero for distributions with the same mean and standard deviation. Therefore, using kurtosis and standard deviation together offers a more sophisticated and nuanced perspective on the distribution of gradients. A high kurtosis with a low standard deviation indicates all the gradients becoming stiff, or low kurtosis with a high standard deviation indicates a broader spread of gradients. This approach captures more complex patterns in the gradients and provides a more fine-grained weighting scheme. The formula for determining weights is given by
(12)$$\begin{array}{cc} {\hat{\lambda }}_{k}\left(\tau \right) & =\frac{\mathrm{kurt}\{\nabla_{\theta^{\mathrm{sh}}}L_{k}\left(\tau \right)\}}{\mathrm{std}\{\nabla_{\theta^{\mathrm{sh}}}L_{k}\left(\tau \right)\}}. \end{array}$$

The operator kurt{·} denotes the calculation of the kurtosis value of the distribution of gradients of a loss term.

## 4. Experiments and Results

We apply the proposed schemes in the previous section to PINNs to assess their capabilities. We analyze their performance when compared to each other. To highlight the capabilities of the proposed schemes to improve PINNs, we solve the 2D Poisson’s Equation and the Klein–Gordon Equation. These cases are used in the literature as a benchmark since classical formulation PINNs struggle to predict the correct solution for these equations. We use the PyTorch [27] package for constructing the neural network and loss functions to approximate the solution of the differential equations. The metrics used to validate the prediction of PINN is the relative $L^{2}$ error between the prediction and the ground truth solution. The code will be available on GitHub (https://github.com/cvjena/GradStats4PINNs, accessed on 19 October 2023). We considered six weighting schemes for determining the weights for the loss function, given by (Equation 7). We used the following notation shown in Table 1.

The metric used to quantify the performance of PINN is the $L^{2}$ error between the true solution (denoted as $u_{true}$) and the predicted solution by the PINN (denoted by $u_{PINN}$). The formula is given by
(13)$$L^{2}\;\mathrm{error}=\frac{\sqrt{|u_{PINN}-u_{true}{|}^{2}}}{\sqrt{|u_{true}{|}^{2}}}.$$

### 4.1. The 2D Poisson’s Equation

For the initial set of experiments, we solve the 2D Poisson’s Equation using PINNs. It is a ubiquitous equation in many fields like gravity, electrostatics, and fluid mechanics. This equation, Helmholtz, and the wave equation belong to the same family and are challenging cases where classical PINNs are known to struggle [11,16,17]. The 2D Poisson’s Equation in x and y is given by
(14)$$\frac{\partial^{2}u}{\partial x^{2}}+\frac{\partial^{2}u}{\partial y^{2}}=-2k^{2}\cos\left(kx\right)\sin\left(ky\right)\;x\in \left[0,1\right],\;y\in \left[0,1\right],$$
where ***k*** is the wave number that determines the frequency of the solution ***u***(***x***,***y***). The complexity of the solution increases with increasing ***k***. In [22], they show that the classical implementation performs worse with increasing ***k***. Therefore, we chose k=6π to exemplify the usefulness of weighting schemes. The boundary conditions can be determined by taking samples along the boundaries using an analytical solution. This equation has an analytical solution given by
(15)u(x,y)=cos(kx)sin(ky).

We aim to measure the performance of the models as mentioned earlier in Table 1 in solving this benchmark problem for different architectures of a feed-forward neural network. We achieve this by varying the number of hidden layers and neurons per layer. We use a Sigmoid Linear Unit (Swish) as an activation function for all the architectures, as prescribed in [28]. Following the related works [18,22], we uniformly sample 2500 points inside the domain ($N_{r}=2500$) and 500 points along the boundaries. ($N_{bc}=500$). We train the neural networks with the most commonly used Adam optimizer (with default hyperparameters prescribed by [29]) for 40,000 epochs. We chose the Adam optimizer because of its robustness and it is proven to provide efficient and adaptive updates to network weights [29].

The relative $L^{2}$ errors for different architectures are in Table 2. We see that the performance of the PINN is vastly improved by the weighting schemes over classic formulation (W1). The general trend is that more layers and units lead to a better result. This makes sense because a larger network means more parameters are used in training. The proposed weighting schemes W5 or W6 for every architecture show the best performance. We observe that the lowest $L^{2}$ error between the ground truth and prediction is obtained when we employ W6 for a neural network with five layers and 100 units each. The smallest neural network containing three layers and 30 units each gives the largest $L^{2}$ errors between the prediction and ground truth. These results demonstrate some interesting implications for the proposed weighting schemes. W5 and W6 outperform the existing weighting schemes. W4 does not outperform W3, and its behavior is similar to W2. This is because the sum of the mean and the standard deviation is almost equal to the mean, which makes the behavior of W4 and W2 similar. We can also see that W5 favors smaller architectures to reach a better prediction, and as the size increases, W6 performs better.

We also provide the evolution of $L^{2}$ error during the training of PINN when using all the different schemes in Figure 3. The graph shows how the weighting schemes avoid the erroneous solution of W1 to reach a better prediction, and W6 has the best prediction of all the weighting schemes, followed by W5 (Table 3). We provide visualizations of the predictions of PINNs for one particular architecture of five hidden layers and 50 units per layer in the Figure 4. When no weighting is used, the PINN fails to capture the ground truth solution, giving rise to a large $L^{2}$ error. This can be visually seen in the large areas of discrepancies shown in the Figure 4. Using W2 shows considerable improvement. The $L^{2}$ error decreases a lot in this case when we compare it with W1. Still, the solution has visible discrepancies, especially near the boundaries. W4 has a slightly better fit than W2 and still has discrepancies around the boundaries. W3, W5, and W6 give the least $L^{2}$ errors and thus perform the best in this case. W6 outperforms W3 and W5, having the least $L^{2}$ error of these three.

To show the robustness of the proposed weighting schemes, we repeat the same experiment by adding noise to the observations used for boundary conditions. These experiments show the effectiveness of the proposed schemes against noisy observations. We add a random Gaussian noise (random values generated from mean = 0 and standard deviation = 0.01) to the boundary data while the remaining training procedure, architecture, and hyperparameters remain the same as before. The results are tabulated in Table 4. We observe that although the additional noise has an effect on overall performance, the proposed weighting schemes are quite robust and still have better performance than W1 and W2.

### 4.2. Klein–Gordon Equation

For the next set of experiments, we consider the Klein–Gordon Equation. It is given in (Equation 16). The Klein–Gordon Equation is a relativistic wave equation with important applications in various fields, including quantum mechanics, quantum field theory, optics, acoustics, and classical mechanics. It has time-dependent non-linear terms, posing an interesting challenge for PINNs:(16)$$\frac{\partial^{2}u}{\partial t^{2}}+\alpha \frac{\partial^{2}u}{\partial x^{2}}+\beta u+\gamma u^{k}=f\left(x,t\right),\;x\in \left[0,1\right],\;t\in \left[0,1\right].$$

The initial and Dirichlet boundary conditions are given in (Equation 17):(17)$$u\left(x,0\right)=x,\;\frac{\partial u}{\partial t}\left(x,0\right)=0u\left(0,t\right)=0,\;u\left(1,t\right)=\cos\left(5\pi t\right)+t^{3}.$$

We choose the parameters to be α=−1, β=0, γ=1, andk=3. We have the ground truth solution and cubic non-linearity for this set of parameters. The ground truth solution for this equation with boundary and initial conditions and given parameters is
(18)$$u\left(x,t\right)=x\cos\left(5\pi t\right)+{\left(xt\right)}^{3}.$$

We estimate the consistent forcing term f(x,t) as prescribed by [17]. To elucidate the capabilities of proposed schemes to handle non-linear time-dependent terms, we proceed by training a PINN to solve (Equation 16) along with its BCs and ICs and compare the predicted solution with the ground truth given by (Equation 18). We use a feed-forward neural network of 7 layers and 50 units, each with a hyperbolic tangent activation function. The number of collocation points in the domain($N_{r}$) is 1000, with 300 points along the boundaries ($N_{bc}=300$). The training uses an Adam optimizer with default parameters for 40,000 steps. The resulting $L^{2}$ errors of the approximated solution of the Klein–Gordon Equation with ground truth are given in Table 5.

Visualizations of approximated solutions along with ground truth and point-wise errors are given in Figure 5.

W1 failed to capture the solution, leading to a high $L^{2}$ error. For this particular problem, all the weighting schemes (W2–W6) have improved over W1, and the fit is also very good. We see that W6 performs the best of all the weighting schemes, having the lowest $L^{2}$ error. This trend is visible in Figure 5. We see significant differences between ground truth and prediction for W1, leading to clearly visible contours in the point-wise error plot. The point-wise errors tend to decrease (the blackness in the plot increases) from W2 to W6, and we observe that W6 has the best fit, following Table 5. This elucidates the effectiveness of the proposed weighting schemes. It has to be noted that even though the solution has improved significantly, the $L^{2}$ error is still higher than in Poisson’s Equation. This is due to the highly non-linear time-dependent terms, which can be further improved by using advanced architectures [19] or adaptive activation functions [20]. The evolution of the $L^{2}$ error during training is shown in Figure 6. The figure shows that weighting schemes reach a better solution, and W6 outperforms them all.

## 5. Discussion and Conclusions

PINNs can potentially include knowledge in the form of differential equations in neural networks, improving the modeling and simulations of many physical systems. This potential is yet to be realized as PINNs often struggle to approximate solutions in many cases. We outlined various works in the literature that show this struggle is due to the imbalanced training of the multi-objective loss function used to train the PINN.

In this paper, we propose using statistics of individual backpropagated gradients of loss terms to assign weights dynamically. We proposed a collection of weighting schemes based on gradient statistics to improve the training of PINNs. These schemes follow a common principle of increasing the weight of a loss term if its distribution of backpropagated gradients tends to approach zero. This approach prevents the loss term from vanishing and enables the PINN to converge to a better optimum. We introduce three novel weighting schemes extending the mean-based [17] and standard deviation-based [22] weighting schemes. We use the combinations (sum and product) of mean and standard deviation as weighting schemes to leverage the information about the gradients’ variability and central tendency. Lastly, we use the kurtosis of distribution to have more nuanced information about the peakedness of gradients around the mean to formulate a kurtosis and standard deviation-based weighting scheme (W6).

The proposed schemes (two from the literature, three novel) were rigorously tested along with the classical formulation qualitatively and quantitatively by solving 2D Poisson’s and Klein–Gordon’s Equations. The metric relative $L^{2}$ error results showed that weighting drastically improved the performance of the PINN. The results proved that the weighting schemes W5 and W6 (namely, product of mean and standard deviation-based and kurtosis–standard deviation-based) outperform existing schemes, with W6 being the most robust. W4 does not perform considerably compared to W5 and W6 because the sum of the mean and standard deviation is almost equal to the mean. Thus, this set of weighting schemes using only backpropagated gradient statistics provides an easy and efficient means to significantly improve the training of PINNs. We have also tested the robustness of the proposed schemes for noisy observations.

The main limitation of our proposed schemes is that they are empirical in nature. These, albeit providing an efficient solution, they do not inform on how actual non-convex optimization is being affected by weighting different loss terms. Hence, there will be cases where these schemes may not be as efficient as other methods. This needs further investigation. While this paper primarily takes an empirical approach, further investigation into the impact of these schemes on the training dynamics would be an exciting avenue for future research. Understanding the underlying mechanisms and analyzing how these schemes affect learning could provide valuable insights. The future of PINNs holds exciting prospects, and the field of deep learning in physics continues to evolve, promising advancements in various scientific domains.

## Figures and Tables

**Figure 1 sensors-23-08665-f001:**
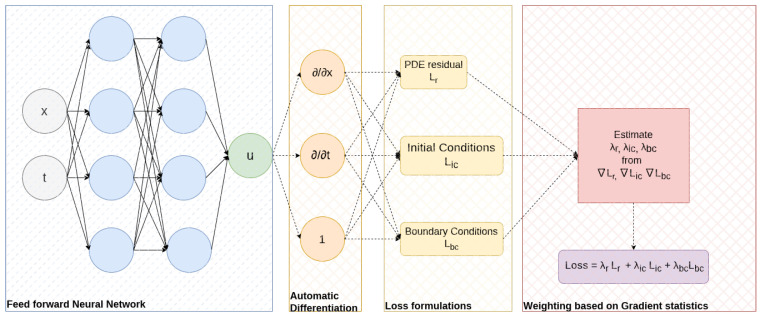
Schematic description of weighting loss terms based on gradient statistics in PINNs. The loss function is formulated by weights derived as a function of backpropagated gradients of loss functions.

**Figure 2 sensors-23-08665-f002:**
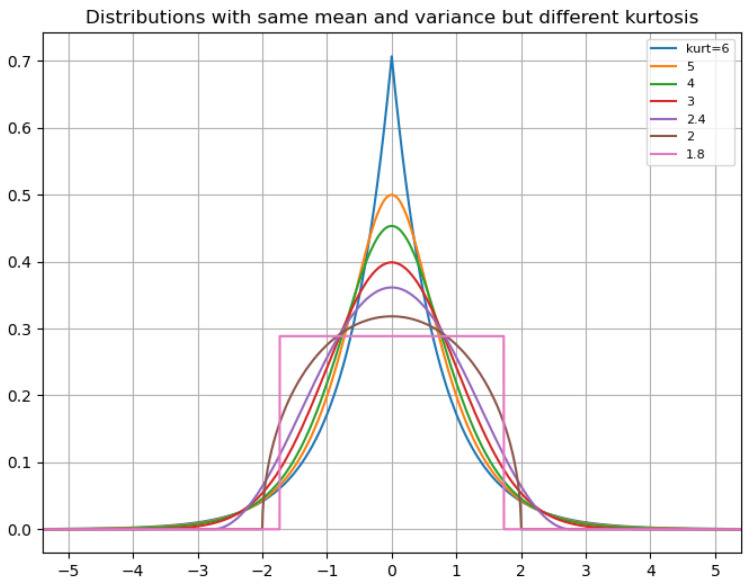
The plot shows different kurtosis values of curves having the same mean and standard deviation, and that a peaked curve has high kurtosis.

**Figure 3 sensors-23-08665-f003:**
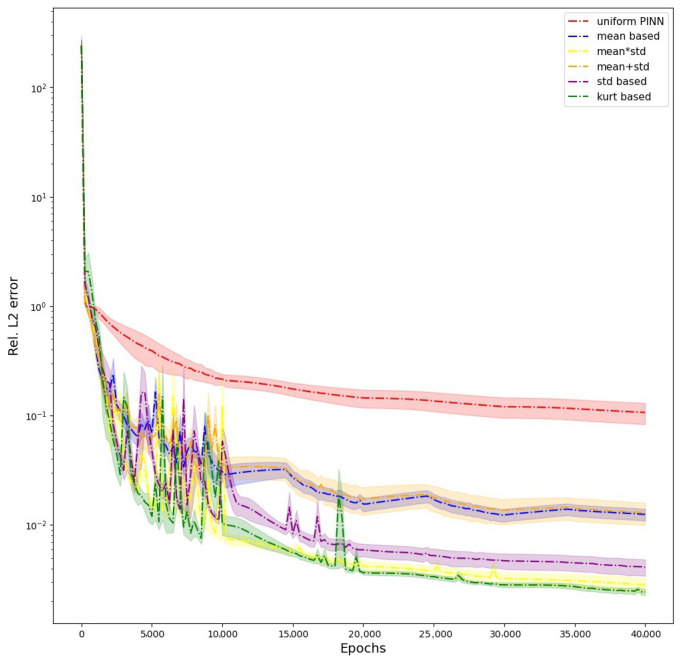
Poisson Equation: Graph showing evolution of $L^{2}$ errors during the training for all the weighting schemes.

**Figure 4 sensors-23-08665-f004:**
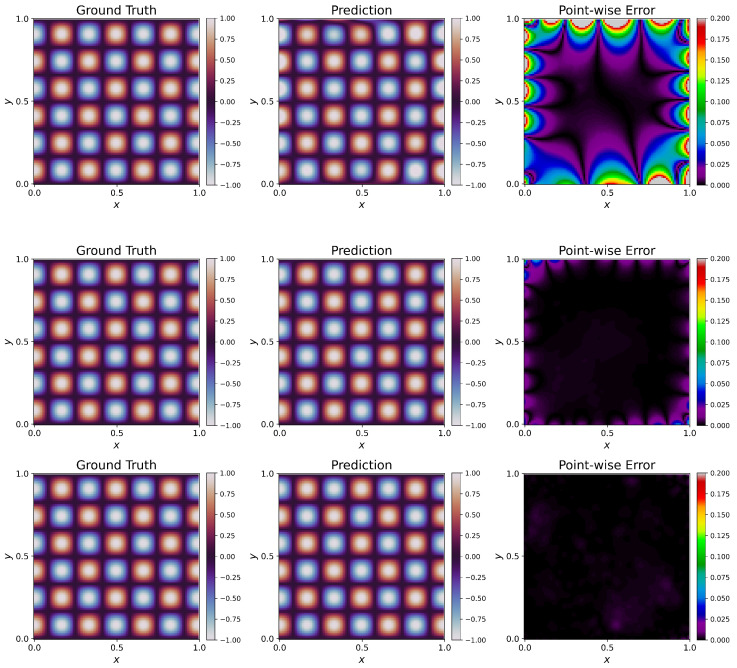
Poisson Equation: Prediction of PINNs using schemes W1–W6 (**top**–**bottom**). Prediction (**middle**) of W1 fails to capture the ground truth (**left**), resulting in high point-wise errors (**right**). W2 and W4 showed a better fit but still have observable point-wise discrepancies, especially along the boundaries. W3, W5, and W6 have shown the best fit and least order of point-wise errors, with W6 having the least (best performing) $L^{2}$ error among them.

**Figure 5 sensors-23-08665-f005:**
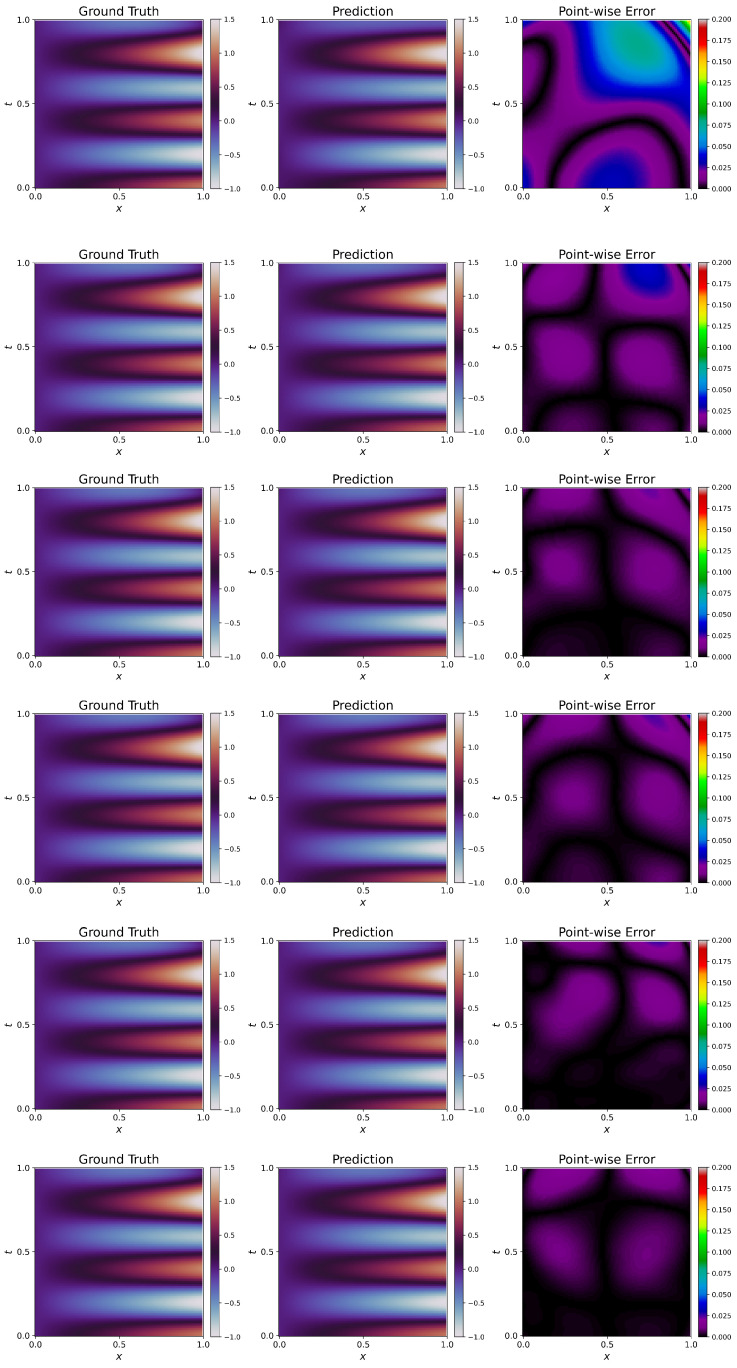
Klein–Gordon Equation: Prediction of PINNs using schemes W1–W6 (**top**–**bottom**). Prediction (**middle**) of W1 fails to capture the ground truth (**left**), resulting in high point-wise errors (**right**). Prediction clearly improved from W1 to W6.

**Figure 6 sensors-23-08665-f006:**
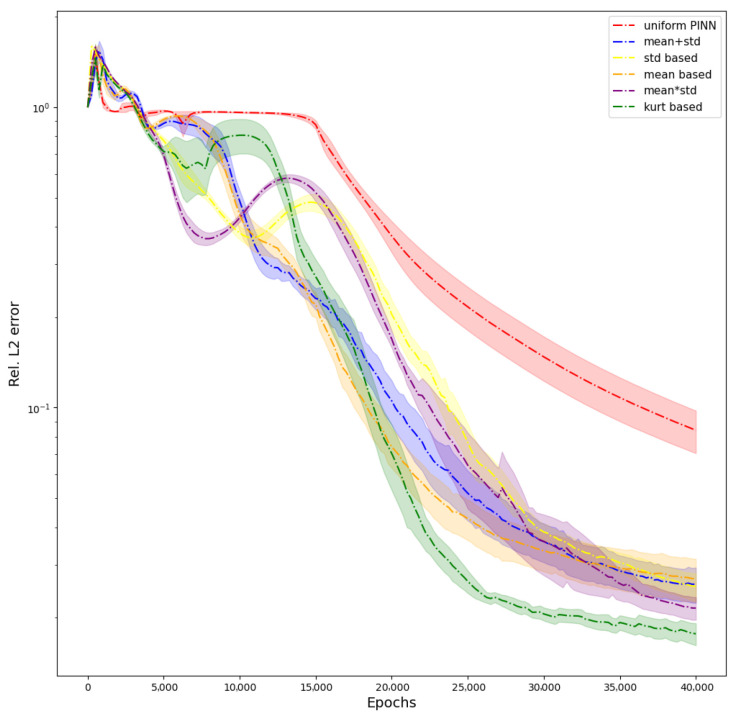
Solving the Klein–Gordon Equation using a PINN: evolution of $L^{2}$ error during training.

**Table 1 sensors-23-08665-t001:** Symbols and descriptions of different weighting schemes.

Symbol	Description
W1	Uniform weighting of all loss terms, the original implementation of Raissi et al. [2]
W2	Mean-based weighting scheme, original implementation of Wang et al. [17], given by (Equation 8)
W3	Standard deviation-based weighting scheme, the original implementation of Maddu et al. [22], given by (Equation 9)
W4	Weighting by the sum of mean and std, given by (Equation 10).
W5	Weighting by the product of mean and std, given by (Equation 11).
W6	Weighting by the ratio of kurtosis and std, given by (Equation 12).

**Table 2 sensors-23-08665-t002:** Table of $L^{2}$ errors between ground truth and prediction solutions for different layers and units, using different weighting schemes for solving the 2D Poisson’s Equation ^1^. The lowest $L^{2}$ error for each set of architecture is shown and highlighted in bold.

Units/Layers	W1	W2	W3	W4	W5	W6
100/5	0.0871	0.0071	0.0031	0.0065	0.0028	**0.0019**
50/5	0.0938	0.0194	0.0056	0.01511	0.0041	**0.0031**
30/5	0.2049	0.0274	0.0086	0.0273	**0.0058**	0.0065
100/3	0.0908	0.0162	0.0060	0.0160	0.0059	**0.0048**
50/3	0.1337	0.0308	0.0072	0.028	**0.0059**	0.0061
30/3	0.2773	0.0529	0.0208	0.0482	0.0124	**0.0103**

^1^ Using Adam optimizer, SiLU activation with default parameters. $L^{2}$ error is the mean value from 10 independent trials. Loss Weights are updated for every five epochs.

**Table 3 sensors-23-08665-t003:** Table of $L^{2}$ errors between ground truth and prediction solutions using different weighting schemes for solving 2D Poisson’s Equation ^1^. The lowest $L^{2}$ error is shown and highlighted in bold.

Scheme	W1	W2	W3	W4	W5	W6
$L^{2}$ error	0.0938 ± 0.0163	0.0194 ± 0.0036	0.0056 ± 0.0015	0.0151 ± 0.0060	0.0041 ± 0.0013	**0.0031** ± **0.0008**

^1^ Using Adam optimizer, SiLU activation with default parameters. L2 error is from 10 independent trials. Loss weights are updated for every 5 epochs.

**Table 4 sensors-23-08665-t004:** Table of $L^{2}$ errors between ground truth and prediction solutions using different weighting schemes for solving 2D Poisson’s Equation ^1^ with added noise. The lowest $L^{2}$ error is shown and highlighted in bold.

Scheme	W1	W2	W3	W4	W5	W6
$L^{2}$ error	0.1445 ± 0.0196	0.0231 ± 0.0086	0.0061 ± 0.0019	0.0192 ± 0.0058	0.0050 ± 0.0016	**0.0041** ± **0.0010**

^1^ Using Adam optimizer, SiLU activation with default parameters. L2 error is from 10 independent trials. Loss weights are updated for every 5 epochs.

**Table 5 sensors-23-08665-t005:** Table of $L^{2}$ errors between ground truth and prediction solutions using different weighting schemes for solving the Klein–Gordon Equation ^1^. The lowest $L^{2}$ error is shown and highlighted in bold.

Scheme	W1	W2	W3	W4	W5	W6
$L^{2}$ error	0.1037 ± 0.0272	0.0249 ± 0.0062	0.0248 ± 0.0071	0.0235 ± 0.0088	0.0219± 0.0051	**0.0165** ± **0.0035**

^1^ Using Adam optimizer, hyperbolic tangent activation with default parameters. $L^{2}$ error is from 10 independent trials. Loss weights are updated for every five epochs.

## Data Availability

All the code for experiments and visualizations will be made available at (https://github.com/cvjena/GradStats4PINNs, accessed on 19 October 2023).

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
