# Peer review of "Gradient Statistics-Based Multi-Objective Optimization in Physics-Informed Neural Networks"

_sensors, 2023, doi:10.3390/s23218665_

Round 1

Reviewer 1 Report

In this paper, the authors proposed a collection of weighting schemes based on gradient statistics to improve the training of PINNs, including the combinations (sum and product) of mean, standard deviation, and kurtosis of distribution. Experiments on 2D Poisson’s and Klein-Gordon’s equations. show the validity of the method.

Comment 1: How to show that W6 is robust? Suggest adding simulation experiments containing noise.

Comment 2: What is the design idea behind equation (13)? Why is the combination of kurt and std used for the numerator and denominator? It is recommended that this innovation be explained in more detail, accompanied by some experiments in simple statistics, to highlight the innovation of this paper.

Comment 3: The presentation, typesetting, and equations of the paper need to be revised, including but not limited to the following:

  • Is there a writing error in formula (6)?
  • Wrongly written "where" after a formula, e.g. line 109, 120, 228, 280, etc.
  • It is suggested that basic explanations of the mathematical representations and statistics of some operators such as $\text{std}(\cdot)$ and $\text{kurt}(\cdot)$ be given.
  • The article uses the same weighted average in several places, such as equations (8), (9), (12), and (14), and it is recommended to merge them.
  • It is suggested that drawings be distinguished using different labels rather than colors.

Author Response

Dear Respected Reviewer

We cordially thank you for taking the time and provide such an encouraging and helpful review. We have taken your comments and suggestions and made changes accordingly in the manuscript.

Here, we provide our response to the comments and suggestions:

  1. We completely agree with the comment that experiments with adding noise to the observations will prove the robustness of W6. We think this is an excellent addition. We have accordingly conducted the experiment and the results can be found in lines  346-354.
  2. The main idea behind the formulation of (13) is that a higher kurtosis value indicates gradients peaked around zero even for a constant standard deviation. Hence, we use kurtosis as the numerator and standard deviation as the denominator to have a nuanced measure of weight and improvement over W3. Please find more information in the section 3.4.
  3. We thank you for suggesting extensive changes in the typesetting of equations, grammar, etc. We have taken time to extensively correct such discrepancies and here are some responses:
    1. Yes, there is an error in (6) and we have corrected that.
    2. Thank you for pointing this out, we have changed the wrong implementation of "where" immediately after an equation.
    3. We explained the basic definitions of operators like $\text{std}(\cdot)$ and $\text{kurt}(\cdot)$ as suggested. We also explained how these statistics are calculated in lines 221-225.
    4. We agree with the suggestion that using the moving averages formula multiple times is causing redundancy. We made sure to define it only once and explicitly state that it remains the same for all schemes.
    5. We found that using different colors for each scheme (In Graphs) and representing the labels in the legend was more informative and less cumbersome.

We sincerely found these suggestions and comments helpful in improving the manuscript. 

Yours Sincerely,

Authors

Reviewer 2 Report

The presented submission is quite an interesting research on a hot topic of PINN.

The overall impression from the presented text is quite satisfactory. The text itself is well-written, well-formatted, and well-structured. The authors’ logic is clear and somewhat easy to follow. The insight obtained after reading is somewhat interesting.

I see only the following minor issues with the submission.

Please, explicitly state the contributions of the submission at the end of the introduction section (preferably in a form of an itemized list). Otherwise, at a first glance, it is unclear.

It is preferable to expand the state-of-the-art part making the description of the existing works more thorough. Up to now this part of the text is very curt.

Please, do not refer to the equation with the word “Equation”.

Please, see some problems with formatting in (6).

Some abbreviations are not introduced, for example, IC, BC in line 161.

The authors do not explain why they use ADAM as an assumed optimizer. It is not the single possible choice. Moreover, all the assumptions must be explicitly stated in the text.

Author Response

Dear Respected Reviewer

We cordially thank you for taking the time and provide such an encouraging and helpful review. We have taken your comments and suggestions and made changes accordingly in the manuscript.

Here, we provide our response to the comments and suggestions:

  1. We completely agree that providing an itemized list of contributions at the end of the introduction makes it more clear. We have added it in the introduction from the line 107.
  2. We have provided state-of-art literature in lines 67-70 and also in 101-104. Additionally, we have explained in detail state of the art schemes of W2 and W3 in sections 3.1 and 3.2. However,  We agree that it is a good suggestion to add more information on them and we have accordingly done it.
  3. Yes, thank you for this suggestion, we have changed the references to the equation to make it correct now. The changes are applied to the entire manuscript.
  4. Thank you for pointing out the formatting errors in (6), we have corrected them.
  5. We have added an explanation as to why we chose Adam in lines 315-318. Adam was chosen owing to its momentum characteristics and demonstrated robustness. Adam is generally used as a default optimizer in many cases of deep learning.

We sincerely found these suggestions and comments helpful in improving the manuscript. 

Yours sincerely,

Authors

Reviewer 3 Report

Overall, the paper discusses an interesting approach to improve the training of Physics Informed Neural Networks (PINNs) by introducing advanced gradient statistics-based weighting schemes. The use of PINNs to solve complex non-linear systems involving differential equations is a promising area of research, and addressing challenges in optimizing the loss function is important for their practical application. Here are some technical comments and suggestions:

1.      The introduction provides a good overview of the motivation behind using PINNs for solving complex systems. However, it would be helpful to include a brief explanation or definition of PINNs for readers who may not be familiar with the term.

2.      It would be beneficial to explicitly state the specific challenges in optimizing the loss function for PINNs. This will help readers understand the exact problem you are addressing.

3.      Given that this work deals with solving complex systems, it is important to ensure mathematical rigor throughout the paper. Explain the mathematical foundations of PINNs, including the partial differential equations involved, in more detail. This will help readers grasp the context better.

4.      Provide a clear and detailed explanation of the proposed advanced gradient statistics-based weighting schemes. Explain how these schemes are calculated, how they differ from existing approaches, and why they are expected to improve PINN performance. Including equations and algorithmic details will be helpful.

5.      The paper mentions a qualitative and quantitative comparison of the weighting schemes on 2D Poisson's and Klein-Gordon's equations. Ensure that the experimental setup, including the choice of datasets, hyperparameters, and evaluation metrics, is clearly explained. Provide more details about the results, including any statistical significance testing if applicable.

6.      It is important to discuss the limitations of the proposed approach. Are there scenarios or types of problems where these weighting schemes might not work as effectively? Acknowledging limitations can help readers understand the scope and applicability of the method.

7.      Discuss the practical implications of the proposed weighting schemes. Are there real-world problems or industries where these schemes could have a significant impact? Providing some examples or case studies would be beneficial.

8.      If there are existing approaches or techniques for improving PINN training, it would be useful to compare the proposed weighting schemes with these methods. Highlight the advantages and disadvantages of each approach.

9.      Summarize the key findings and contributions of the paper in the conclusion section. Emphasize how the proposed weighting schemes advance the state-of-the-art in PINN training.

Author Response

Dear Respected Reviewer

We cordially thank you for taking the time and provide such an encouraging and helpful review. We have taken your comments and suggestions and made changes accordingly in the manuscript.

Here, we provide our response to the comments and suggestions:

  1. We completely agree that it is a good idea to add a basic introduction to PINN for readers who might not be familiar with the term. We have included it introduction section in lines 37-40.
  2. We have explained the challenges with optimizing the loss function in section 2.2. However, we have also added specific challenges as suggested. This can be found in the introduction in lines 61-66.
  3. Thank you for this suggestion. We have taken extensive care to maintain mathematical rigor and explain the terms throughout the paper. For example, we have added the definition of metric used in lines 295-299.
  4. Thank you for this suggestion. We have tried to provide a clear explanation of all the schemes, along with intuition and formulas in section 3. As suggested, We have made some additions to make it more clear and detailed.
  5. We have taken in the suggestion to ensure the inclusion of detailed information about hyperparameters, metrics, etc. To this end, we have made several additions (for example 315-320 explaining optimizer, 346-354 to include noisy observations to point out a few.)
  6. We completely agree with the comment that it is important to discuss the limitations of the current work and cases where this might not be effective. As suggested, we have added limitations stating how this approach is empirical and doesn't provide information about the underlying non-convex optimization. Thus, this approach might not be suitable for some other applications. (lines 415-418)
  7. We have included practical applications of equations used in real-world scenarios. For example: Poisson's Equation finds applications in Geoscience, gravity, etc. (lines 299-302). With regard to implications, this is a step in the right direction in integrating physics knowledge into machine/deep learning but commenting on real-world impact is not yet possible, since the domain is still developing.
  8. The existing approaches to using gradient statistics include W1, W2, and W3. We have provided detailed comparisons with proposed methods and how they will provide improvements to them. We have also provided other approaches with their limitations in lines 61-70 and 93-104.
  9. We have added additional information as suggested in the conclusions about the key findings in lines 415-418.

We sincerely found these suggestions and comments helpful in improving the manuscript.

Yours sincerely,

Authors